# Effects of contact with a dog on prefrontal brain activity: A controlled trial

**Rahel Marti**[1,2]*, **Milena Petignat**[1], **Valentine L. Marcar**[2,3,4], **Jan Hattendorf**[5], **Martin Wolf**[4], **Margret Hund-Georgiadis**[2], **Karin Hediger**[1,2,5,6]

**1** Faculty of Psychology, University of Basel, Basel, BS, Switzerland, **2** REHAB Basel, Basel, BS, Switzerland, **3** Institute of Psychology, University of Zurich, Zurich, ZH, Switzerland, **4** Department of Neonatology, University Hospital Zurich, Zurich, ZH, Switzerland, **5** Department of Epidemiology and Public Health, Swiss Tropical and Public Health Institute, Allschwil, BL, Switzerland, **6** Faculty of Psychology, Open University, Heerlen, Limburg, Netherlands

* r.marti@unibas.ch

**Data Availability Statement:** Data is are available from the Harvard Dataverse database at https://doi.org/10.7910/DVN/U57AK0.

## Abstract

### Background

There is a broad range of known effects of animal contact on human mental and physical health. Neurological correlates of human interaction with animals have been sparsely investigated. We investigated changes in frontal brain activity in the presence of and during contact with a dog.

### Methods

Twenty-one healthy individuals each participated in six sessions. In three sessions, participants had contact with a dog, and in three control sessions they interacted with a plush animal. Each session had five two-minute phases with increasing intensity of contact to the dog or plush animal from the first to the fourth phase. We measured oxygenated, deoxygenated, and total hemoglobin and oxygen saturation of the blood in the frontal lobe/frontopolar area with functional near-infrared spectroscopy (SenSmart Model X-100) to assess brain activity.

### Findings

In both conditions, the concentration of oxygenated hemoglobin increased significantly from the first to the fourth phase by 2.78 µmol/l (CI = 2.03–3.53, $p$ < .001). Oxygenated hemoglobin concentration was 0.80 µmol/l higher in the dog condition compared to in the control condition (CI = 0.27–1.33, $p$ = .004). Deoxygenated-hemoglobin concentration, total hemoglobin concentration, and oxygen saturation showed similar patterns.

### Conclusion

Prefrontal brain activation in healthy subjects increased with the rise in interaction closeness with a dog or a plush animal. Moreover, interaction with a dog stimulated more brain activity compared to the control condition, suggesting that interactions with a dog can activate stronger attentional processes and elicit more emotional arousal than interacting with a nonliving stimulus.

**Funding:** This work was supported by the Swiss National Science Foundation under an Ambizione [grant number PZ00P1_174082/1] to K. H. and by the Stiftung pro REHAB Basel. The funders had no role in study design, data collection and analysis, decision to publish, or preparation of the manuscript.

**Competing interests:** I have read the journal's policy and the authors of this manuscript have the following competing interests: M.W. is president of the board and cofounder of OxyPrem AG. R.M., M. P., V.L.M., J.H., M.H.-G., and K.H. have declared that no competing interests exist.

# 1 Introduction

Although the effects of contact with animals on human mental and physical health have received increasing attention [1–5], the neurophysiological correlates of these effects are not yet fully understood [6, 7]. These correlates are, however, highly relevant to understanding the mechanisms underlying human–animal relationship [8–11] and to designing effective animal-assisted interventions. Authors of several studies have reported that positive interaction with a dog reduces stress parameters, such as blood pressure, heart rate, or cortisol level [12–14], and leads to an increase of neurochemicals associated with bonding or affiliation, such as β-endorphin, oxytocin, and prolactin [15–17]. However, the results for these parameters remain inconclusive [13, 18, 19].

Investigations into neurological correlates in the context of human–animal interaction are scarce. Initial studies have investigated neurological reactions to interactions with animals using neuroimaging techniques [20–27]. Most of these studies presented images of animals, whereas only a small number of investigations have addressed the effects of real animals. A positron-emission-tomography (PET) study observed that brain areas associated with stress and sympathetic arousal were less activated in the presence of a familiar dog than in a relaxing condition [23]. Other investigators have observed lateralization with greater activity in the right frontopolar area while petting a horse compared to petting a plush animal, seeing a horse, or seeing a plush animal [24]. Another study measuring hemodynamic response found that participants reacted with activation in the left inferior frontal gyrus while petting a cat [25]. Moreover, children showed higher activity in the prefrontal cortex in an attention task after interacting with a dog than after interacting with a robot dog [26]. Similarly, in a small pilot study, participants had a stronger brain reaction to a live animal than to a mechanical toy animal [27]. While these studies provide first insights into neurological correlates of the human–animal interaction, additional research is needed to understand what happens in different forms of human–animal interactions. The knowledge gained will be crucial for conducting effective animal-assisted interventions [28]. Dogs are the most common animals used in animal-assisted interventions [4, 29, 30]. The aim of this study was to investigate neurological correlates of different forms of human–dog contact in an animal-assisted intervention setting using a strong study design. To ensure that the results would be as valuable as possible for practical application, we investigated the reactions of the participants in an animal-assisted intervention setting in a clinic and involving direct contact and interaction with a dog. This also enabled us to control for different amounts of contact with the dog.

Interacting with an animal is a social situation that is emotionally relevant to most people [7, 31–34]. Several reviews have identified the prefrontal cortex as the key region for different aspects of social cognitive processing, such as theory of mind/mentalizing [35] and understanding self and others [36]. Activity in the prefrontal cortex is thus important for investigating the underlying mechanisms of human–animal interactions.

Our study aimed to investigate brain activation in the prefrontal cortex of healthy subjects with functional near-infrared spectroscopy (fNIRS) in a controlled trial. We compared different forms of interaction with a dog and different forms of interaction with a plush animal. We expected, first, that the increase of closeness in contact with a dog or plush animal would correlate with an increased amount of stimulation and therefore also with increased brain activity. Second, we hypothesized that participants would exhibit higher brain activity in the dog condition compared to the control condition with the plush animal.

## 2 Materials and methods

### 2.1 Study design

The study had a controlled, within-subject crossover design with repeated measurements. Participants were measured during six standardized sessions (1–6) consisting of three sessions with a live dog and three control sessions with a plush animal. The six sessions took place within 2 weeks. The sequence of the conditions within these six sessions was influenced by the presence of the dog and therefore only partly randomized. The study design was approved by the local ethics committee, Ethics Commission Northwest and Central Switzerland (Project ID 2017–00540), and by the Veterinary Office of the Canton of Basel-Stadt, Switzerland (No. 2713) and was registered at clinicaltrials.gov (NCT03341325). The study design followed the Animals (Scientific Procedure) Act 1986, European Directive EU 2010/63, and the guidelines for handling animals in research as outlined by the Association for Studies on Animal Behavior and the Society for Animal Behavior. All sessions were conducted according to the guidelines of the International Association for Human–Animal Interaction Organizations and the Helsinki guidelines [37, 38]. We planned to compare the results of this study with a study population of patients with severe disorders of consciousness in a future trial, so the study design complied with the requirements for measuring a group of patients with severe disorders of consciousness.

### 2.2 Participants

Twenty-one healthy subjects (10 women, 11 men) participated in this study. Participants were over 18 years old and without allergies or phobias toward dogs. They were recruited with flyers at the Faculty of Psychology at the University of Basel and via an advertisement on the university's website. We obtained written informed consent from every participant before the study started. The sample size was determined a priori based on data from a previous study [39] and with regard to the pilot character of this study.

### 2.3 Procedure

The sessions were held in a room at the neurorehabilitation center REHAB Basel in Switzerland from February 2018 until July 2018. During the experiments, the participants sat upright on a Bobath therapy couch. They faced a white wall located at a distance of 1.5 m. The study staff attached two fNIRS sensors to measure oxygen saturation on the participants' foreheads. Three of the six sessions per participant were conducted in the presence of a dog and three with a plush animal (see Fig 1). The participants therefore had a first, second, and third contact with both the dog and the plush animal. All sessions were videotaped, and heart rate and electrodermal activity were recorded. Each session consisted of five 2-minute phases, which were always conducted in a similar way and in the same order in both the dog and plush-animal conditions. Before each phase, the study staff verbally instructed the participant according to a standardized protocol. The first phase served as a baseline where the participant looked straight at the white wall and relaxed (neutral 1). In the next phase, the participant watched a dog or a plush animal from a distance of 1 m (watching). The dog or plush animal was placed or asked to lie on a mat and a blanket on a height-adjustable table. Then the dog lay down next to the participant on the couch or the plush animal was placed on the participant's thigh. The participant could passively feel the animal but was not yet allowed to pet it (feeling). Next, the participant petted the dog or the plush animal (petting). Finally, there was a second neutral phase where the participant again looked at the white wall while the dog or the plush animal was out of sight (neutral 2). Each phase concluded after 2 minutes, and then there was a short

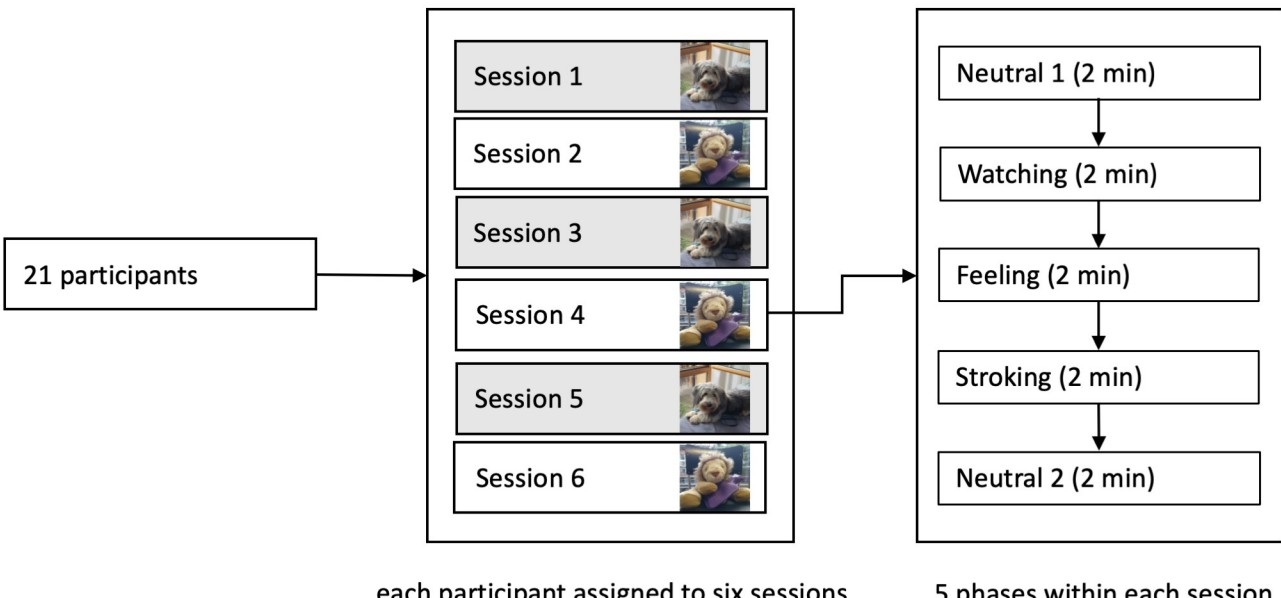

**Fig 1. Study procedure.**

break in which the study staff prepared the room for the next phase. Interactions between each participant and the dog or the plush animal were standardized and comparable regarding the amount of contact.

For every participant, we scheduled three of the sessions in the morning and three in the afternoon to control for time of day. The order of the phases was not counterbalanced because the same design was also used for patients with severe disorders of consciousness. These patients need time and a lot of context to understand a situation. A random order with a sudden increase of contact to the animal would not be ethically justifiable. For the same reason, it was not possible to measure a pretask and posttest baseline for each phase.

## 2.4 Dogs

The dogs participating in the study were used to human contact and trained to work with patients in a hospital setting. The dogs were a female Jack Russel (6 years of age), a female Goldendoodle (4 years of age), and a female Golden Retriever (4 years of age). Each dog participated in a maximum of two sessions in a row. The dogs and their owner were in the room before the sessions started, which enabled them to become acquainted with the room and to feel safe. The dog owner was present throughout the session and was responsible for handling the dog but was instructed not to interact with the participant during the measurements. The dogs were trained to lie silently on the table and beside the participant in contact with the participant's thigh, but they could choose their position themselves. Owners monitored their dogs for signs of stress and predetermined stop criteria. Due to the highly standardized situations

and interaction, the behavior of the dog was comparable between the sessions within and between participants.

## 2.5 Plush animal

For the control sessions, we used a lion plush animal. The plush animal ($58 \times 40 \times 20$ cm) contained in its body a hot water bottle that was filled with warm water before the sessions started to control not only for the sensation of soft fur but also for the body temperature and weight of a dog. We introduced the plush animal to participants as "Leo."

## 2.6 Functional near-infrared spectroscopy

We chose fNIRS to measure the response in the prefrontal cortex as it is particularly suited for investigating the neuronal correlates of such a complex social situation of human–animal interaction. fNIRS has been used as a noninvasive technique to measure brain activity within the context of human–animal interactions [24, 25, 27, 40, 41]. Compared to functional magnetic-resonance imaging (fMRI) or PET, participants are not confined to a scanner but can sit or stand during measurements. This makes the test situation more comparable to clinical situations. fNIRS also has other advantages: there are no disturbing sounds, and the device is easy to handle. fNIRS is a vascular-based neuroimaging technology that measures the oxygen saturation of hemoglobin and changes in total hemoglobin concentration (tHb) based on the characteristic hemoglobin-absorption spectra in the near-infrared range. This technology relies on the well-known tight neurovascular coupling, which induces changes in oxygen saturation and tHb in response to neuronal activity. An increase in oxygenated hemoglobin ($O_2Hb$) in the region of an activated cortical area mirrors increased brain activity [42].

We recorded percent oxygen saturation (%) and tHb (g/dl) in the prefrontal cortex using a Nonin fNIRS device (SenSmart Model X-100). Two sensors of the device (Model 8004CA Sensors–Adhesive) were placed right and left of the midline on the forehead as close to the hairline as possible and then attached with an adjustable band. This corresponded to locations F1, F3, F2, and F4 on the frontopolar area according to the international 10–20 system and to the Brodmann areas 9, 10, and 46. The wavelength of the infrared light was 730, 760, 810, and 880 nm, and measurements were recorded at a frequency of 0.25 Hz. After recording, data were transferred to a laptop using SenSmart software (version 1.0.1.0).

Within this study, we also measured other physiological endpoints such as heart rate, heart-rate variability, and skin conductance. These data will be published separately.

## 2.7 Data processing and analysis

We converted the data from g/dl to μmol/l based on the molar mass of hemoglobin of 64458 g/mol. We calculated the concentration of $O_2Hb$ and HHb from raw data. To exclude unreliable data due to measurement errors, two raters independently rated plots of the data for reliability. The raters were blinded for the condition. Conflicts were resolved by a third rater (R. M.).

For all included data, we calculated the mean concentration of $O_2Hb$, HHb, and tHb and mean oxygen saturation in each phase. To do so, we cut the data from one session into segments of five 2-minute phases at the markings. The data between the phases was not used. We were interested in changes from phase to phase, so we subtracted the mean of the first phase from each following phase within the same session for each participant.

$O_2Hb$ reflects the neuronal-discharge frequency, while HHb reflects the quantity of recruited neurons [43]. We chose $O_2Hb$ as the primary outcome because $O_2Hb$ more directly reflects task-related cortical activation than does HHb [44]. HHb, tHb, and oxygen saturation served as secondary outcomes. For the primary and secondary outcomes, we conducted

prespecified linear mixed-effect models and used the mean difference as the effect size. Within the models, condition and phase were used as fixed effects, and an intercept for the participant was used as the random effect. We conducted the same models again with visibility of the dog owner as a fixed effect.

We conducted explorative analyses because repetition of contact with the dog or the plush animal seemed to influence the outcome. Within these nonprespecified linear mixed-effect models, condition and contact (first, second, or third contact between participant and dog or plush animal) were used as fixed effects. Moreover, we included an interaction term and an intercept for the participant as the random effect.

We visually checked the normality (q-q plot, histogram of residuals), linearity, and homoscedasticity (residuals vs. fitted plot), and influential outliers (leverage and Cook's distance). Leverage was checked with the R package influence.ME [45]. The significance level was set at .05. All analyses were conducted with R 4.1.0 [46] and R package lme4 [47].

# 3 Results

Of the 21 participants measured between January and July 2018, one participant dropped out after one session. We conducted 119 of the 126 planned sessions (Fig 2). Of these 119 sessions, we excluded data from one channel for 55 sessions and from both channels for 10 sessions due to low data quality (Fig 2). Six of these 10 completely removed datasets originated from one participant who dropped out of the analysis, while the other removed datasets were distributed among different participants. We thus analyzed 108 sessions (53 dog conditions, 55 plush-animal conditions) of 19 participants with at least one of the two channels available.

These 19 participants compromised nine women and 10 men. The mean age was 32.4 years ($SD$ = 12.8) and did not differ between the sexes (estimate = 2.2, CI = −15.4–11.1, $p$ = .732). On average, we analyzed 2.89 control sessions and 2.84 dog sessions per participant. The number of analyzed sessions per participant did not differ between the conditions ($M$ = 2.87, $SD$ = 0.34; estimate = −0.05, CI = −0.18–0.28, $p$ = .642). The first session was significantly more often the dog condition (14/19, $p$ = .025), and the second session was significantly more often the control condition (14/18, $p$ = .025). In sessions three to six, the number of sessions per condition did not differ significantly. In two-thirds of the sessions in the dog condition, the participant could see the dog owner during the measurement. No adverse or unintended effects in participants or in the involved dogs occurred during data collection.

## 3.1 Primary analysis

With increased stimulation, oxygenated hemoglobin ($O_2Hb$) in the prefrontal lobe increased significantly from phase neutral 1 to phase petting by 2.78 μmol/l (CI = 2.03–3.53, $p$ < .001). After removal of the stimulation in phase neutral 2, $O_2Hb$ stayed constant and was still significantly higher compared to phase neutral 1 (estimate = 2.91 μmol/l, CI = 2.16–3.65, $p$ < .001).

$O_2Hb$ was 0.80 μmol/l higher in the presence of the dog compared to in the presence of the plush animal (CI = 0.27–1.33, $p$ = .004). The difference between the conditions was highest in the phase petting (Fig 3A). This result was not influenced by the visibility of the dog owner.

## 3.2 Secondary analysis

**3.2.1 Deoxygenated hemoglobin.** When stimulation increased, deoxygenated hemoglobin (HHb) in the prefrontal lobe decreased significantly from phase neutral 1 to the petting phase by 1.23 μmol/l (CI = −1.75 to −0.72, $p$ = .003). After removal of the stimulation in phase neutral 2, HHb stayed constant and was still significantly lower compared to phase neutral 1 (estimate = −1.20 μmol/l, CI = −1.72 to −0.69, $p$ = .005).

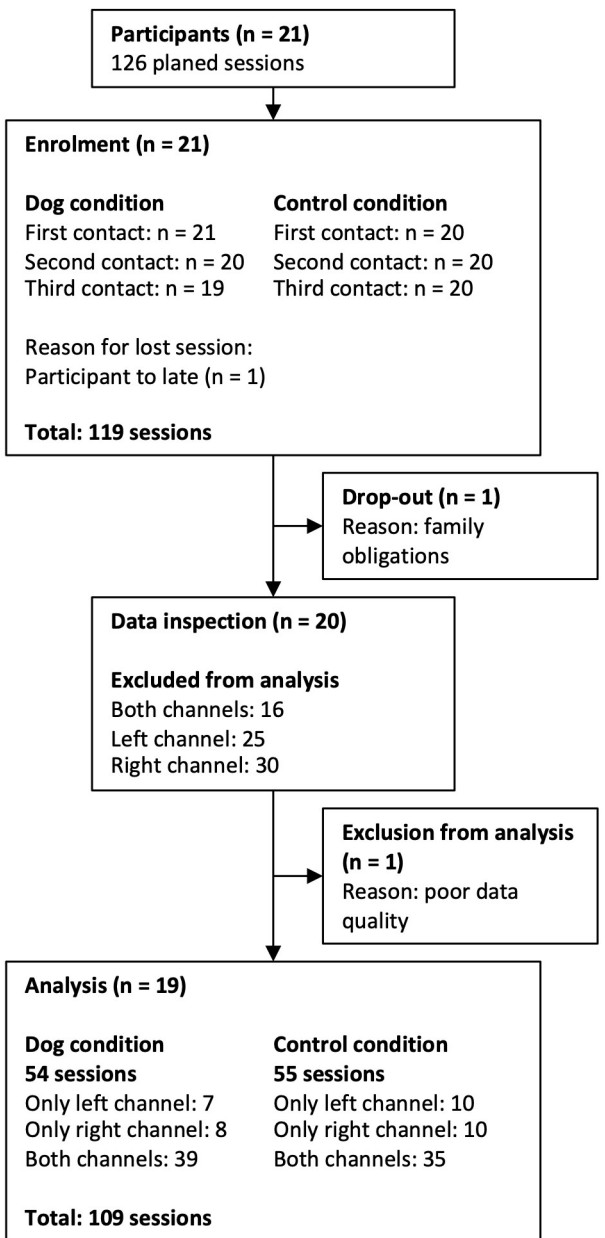

**Fig 2. Flow diagram of the study.**

HHb tended to be lower in the presence of the dog compared to in the presence of the plush animal (estimate = -0.35 μmol/l, CI = −0.71–0.02, $p$ = 0.064). The difference was highest in phase neutral 2 (Fig 3B). This result was not influenced by the visibility of the dog owner.

**3.2.2 Total hemoglobin.** When stimulation increased, total hemoglobin (tHb) in the prefrontal lobe increased significantly from phase neutral 1 to the petting phase by 1.54 μmol/l (CI = 1.08–2.01, $p < .001$). After removal of the stimulation in phase neutral 2, tHb stayed constant and was still significantly higher compared to phase neutral 1 (estimate = 1.70 μmol/l, CI = 1.24–2.17, $p < .001$).

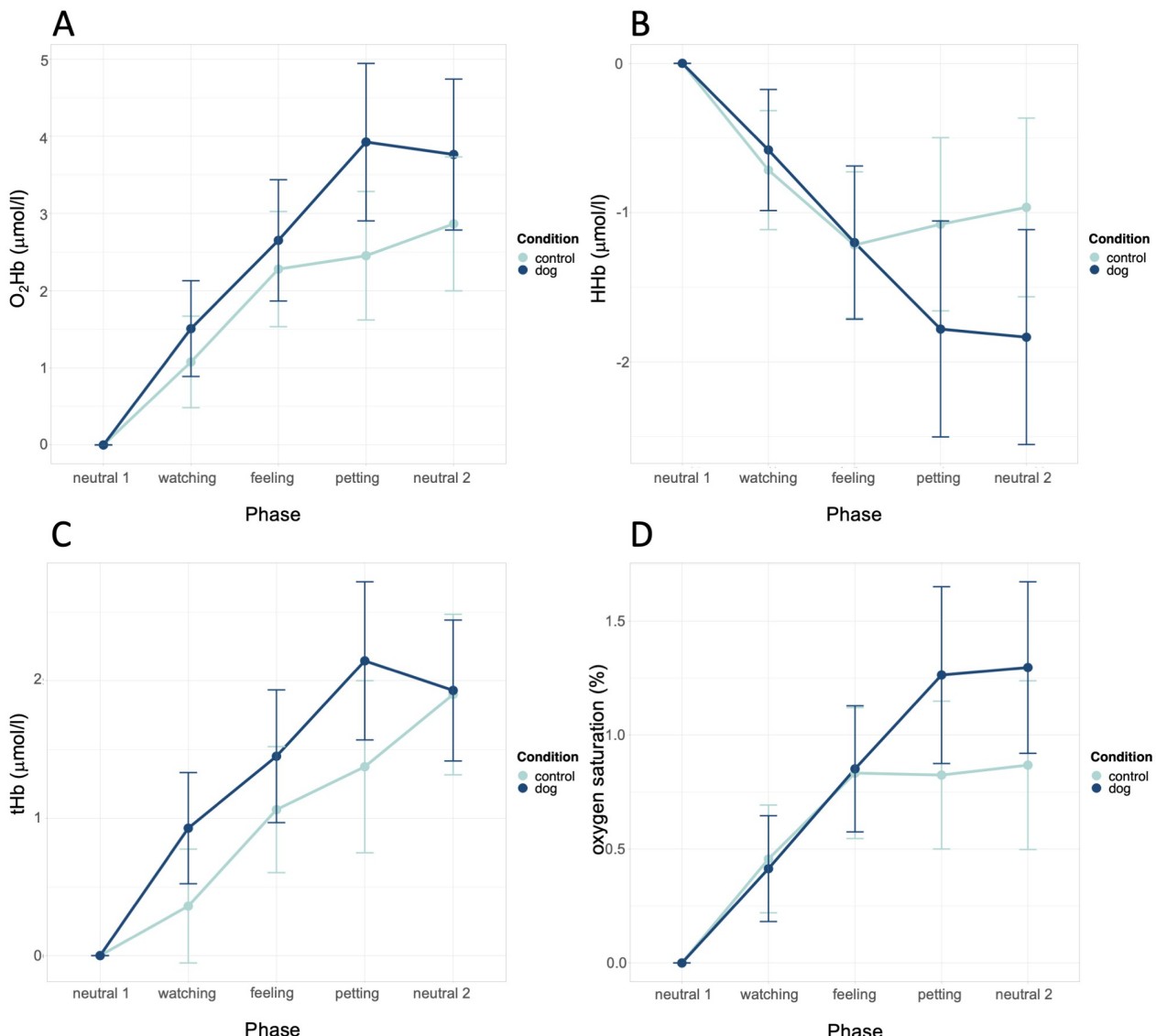

**Fig 3. Effects of condition and phase on O$_2$Hb, HHb, tHb, and oxygen saturation.** (A) O$_2$Hb, (B) HHb, (C) tHb, and (D) oxygen saturation. Error bars denote confidence interval. Data is shown as relative change from phase neutral 1.

The concentration of tHb was significantly higher by 0.45 µmol/l in the presence of the dog compared to in the presence of the plush animal (CI = 0.12–0.78, $p$ = .008). The difference was highest in the petting phase (Fig 3C). In the dog condition, tHb was lower when the participant could see the dog owner than when the dog owner was out of sight (estimate = −0.84, CI = −1.33 to −0.33, $p$ < .001). The results of the other factors in the model, including visibility of the dog owner, remained unchanged.

**3.2.3 Oxygen saturation.** When stimulation increased, oxygen saturation in the prefrontal lobe increased significantly from phase neutral 1 to the petting phase by 0.93% (CI = 0.64–1.22, $p$ < .001). After removal of the stimulation in phase neutral 2, saturation stayed constant and was still significantly higher compared to phase neutral 1 (estimate = 0.97%, CI = 0.68–1.27, $p$ < .001).

Oxygen saturation was significantly higher by 0.21% in the presence of the dog compared to in the presence of the plush animal (CI = 0.00–0.42, $p$ = .047). The difference was highest in phase neutral 2 (Fig 3D). The visibility of the dog owner had no effect.

### 3.3 Explorative analysis

During the first contact (first session), there was no relevant difference in $O_2Hb$ between the dog condition and the plush-animal condition (estimate dog = 2.15 μmol/l, estimate plush animal = 2.60 μmol/l). We observed a significant interaction, which indicates that with repeated contact over time, there was an increasing difference between the dog condition and the plush-animal condition (second contact: $p$ = .001, third contact: $p$ = .023, Table 1, Fig 4A).

There was no relevant difference in HHb between the dog condition and the plush-animal condition during the first contact (dog = −0.98 μmol/l, plush animal = −1.16 μmol/l). We observed a significant interaction between the condition and number of contacts with an effect on HHb in the second contact but not in the third (second contact: $p$ = .002, third contact: $p$ = .695, Table 1, Fig 4B).

During the first contact, there was no relevant difference in tHb between the dog condition and the plush-animal condition (dog = 1.17 μmol/l, plush animal = 1.44 μmol/l). We observed a significant interaction effect on tHb, which indicates that the difference between the dog condition and the plush-animal condition increased with repeated contact over time (second contact: $p$ = .053, third contact: $p$ = .001, Table 1, Fig 4C).

There was no relevant difference in oxygen saturation between the dog condition and the plush-animal condition during the first contact (dog = 0.77%, plush animal = 0.80%). We observed a significant interaction between the condition and number of contacts with an effect on oxygen saturation in the second contact but not in the third (second contact: $p$ = .010, third contact: $p$ = .823, Table 1, Fig 4D).

## 4 Discussion

This study compared the prefrontal brain activity of healthy adults during contact with a dog and contact with a plush animal. Prefrontal activity increased with increased intensity of contact with a dog or a plush animal. This confirms our first hypothesis that more stimulation correlates with higher brain activity. It also corroborates previous studies linking closer contact with animals or control stimuli with increased frontal brain activation [24, 25, 27].

The participants had higher prefrontal brain activity when they interacted with a dog than when they interacted with a plush animal. This confirms our second hypothesis. In the presence of the dog, $O_2Hb$, tHb, and oxygen saturation were significantly higher while HHb tended to be lower compared to the control condition. This pattern indicates increased oxygen consummation in prefrontal areas and thus higher brain activation in the presence of a dog [48, 49]. This result is in line with previous studies. An fNIRS pilot study with patients in a minimally conscious state and healthy controls found that three of four participants showed a higher hemodynamic response when stroking a live animal (dog, rabbit, or guinea pig) compared to stroking a mechanical toy [27]. Children who underwent a 20-min session with a therapy dog after surgery showed faster electroencephalogram diffuse beta activity, while children in the control group who received standard postoperative care showed no beta activity [41]. The passive infrared hemoencepahlography signal of children who performed an attention test was significantly higher after the interaction with a real dog compared to after the interaction with a robotic dog [26].

**Table 1. Marginal effects of condition by number of contacts.**

|  | Estimate | 95% CI Lower limit | Upper limit |
|---|---|---|---|
| **O$_2$Hb** |  |  |  |
| **Dog condition** |  |  |  |
| First contact | 2.15 | 1.35 | 2.95 |
| Second contact | 3.43 | 2.62 | 4.25 |
| Third contact | 3.36 | 2.52 | 4.19 |
| **Plush-animal condition** |  |  |  |
| First contact | 2.60 | 1.78 | 3.41 |
| Second contact | 1.62 | 0.81 | 2.44 |
| Third contact | 2.25 | 1.45 | 3.05 |
| **HHb** |  |  |  |
| **Dog condition** |  |  |  |
| First contact | −0.98 | −1.53 | −0.43 |
| Second contact | −1.70 | −2.26 | −1.14 |
| Third contact | −1.31 | −1.89 | −0.74 |
| **Plush-animal condition** |  |  |  |
| First contact | −1.16 | −1.72 | −0.60 |
| Second contact | −0.44 | −1.00 | 0.12 |
| Third contact | −1.32 | −1.87 | −0.76 |
| **tHb** |  |  |  |
| **Dog condition** |  |  |  |
| First contact | 1.17 | 0.66 | 1.68 |
| Second contact | 1.74 | 1.22 | 2.26 |
| Third contact | 2.04 | 1.50 | 2.57 |
| **Plush-animal condition** |  |  |  |
| First contact | 1.44 | 0.92 | 1.96 |
| Second contact | 1.18 | 0.66 | 1.70 |
| Third contact | 0.94 | 0.42 | 1.45 |
| **Oxygen saturation** |  |  |  |
| **Dog condition** |  |  |  |
| First contact | 0.77 | 0.47 | 1.08 |
| Second contact | 1.10 | 0.78 | 1.41 |
| Third contact | 0.98 | 0.66 | 1.30 |
| **Plush-animal condition** |  |  |  |
| First contact | 0.80 | 0.49 | 1.11 |
| Second contact | 0.45 | 0.13 | 0.76 |
| Third contact | 0.95 | 0.64 | 1.26 |

Marginal effects were estimated by condition and contact number, and an intercept for participant as random effect.

## 4.1 Comparison with other studies

We found that prefrontal brain activity increased with a rise in the intensity of contact with a dog or a plush animal. From watching the animal to feeling it passively to actively petting the animal, the interactional closeness increased and, with it the intensity of stimulation as well as the number of senses involved. This led to an increase in brain activation. We detected the same pattern in a pilot study with a similar study design and comparable forms of contact to an animal [27]. In line with this, another study revealed higher frontopolar activity when

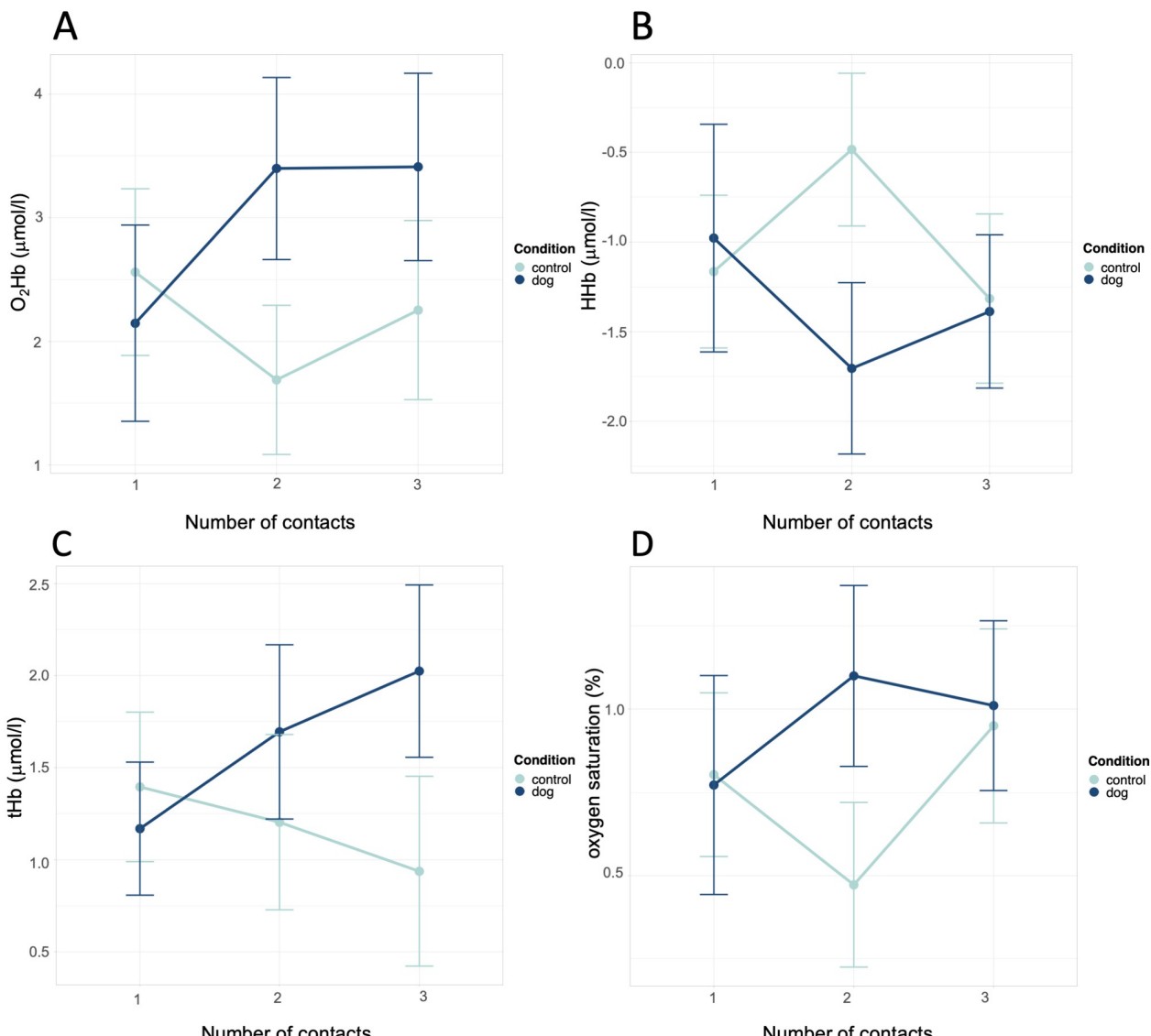

**Fig 4. Effects of condition and number of contacts on O₂Hb, HHb, tHb, and oxygen saturation.** (A) $O_2Hb$, (B) HHb, (C) tHb, and (D) oxygen saturation. Error bars denote confidence intervals. Data are shown as relative change from phase neutral 1. The data for phase neutral 1 are not included in the presented means.

participants stroked a plush animal or a miniature horse compared to just seeing them [24]. Moreover, stroking a cat stimulated higher activation of the inferior frontal gyrus compared to just touching a cat [25].

We observed clear differences in brain activity in the presence of the dog compared to the plush animal. This contrasts with a study reporting that healthy participants had similar activation patterns of the inferior frontal gyrus when petting a cat or a plush animal [25]. That study also noted that female and male participants showed different activation patterns. A PET study observed deactivation in the left middle frontal gyrus, the right fusiform gyrus, the left putamen, and the thalamus in healthy participants during the presence of a familiar dog compared to a resting condition [23]. The authors suggested that this deactivation signaled a reduction in

emotional stress induced by the presence of the familiar dog. These results cannot be directly compared with our results, because fNIRS cannot reach areas like the putamen or the thalamus. Nevertheless, the tasks in our design might have been more activating and our imaging technology less stressful.

Other studies identified lateralized activation patterns in frontal areas during petting a horse or a cat compared to a plush animal [24, 25]. For example, participants exhibited lateralization in the right frontopolar cortex while petting a real horse compared to no lateralization while petting a plush horse[24]. The authors attributed the lateralized activity to differences in function of the left and right frontal regions. We did not test for lateralization in the present study, but visual inspection of our data does not suggest lateralization. However, future studies should address the possibility of lateralization.

Summing up, the current literature indicates that frontal brain activation patterns in humans correlate with the level of interaction with animals. Our results show that this is also the case with a live dog compared to a plush animal and that the intensity of interaction is relevant. Looking at a dog correlates with the lowest frontal activity, while passive contact with more and active stroking correlates with the highest frontal activity.

## 4.2 Brain activity across sessions

In the second neutral phase, brain activation did not return to the level of the first neutral phase. We assume that activation persisted in both conditions and did not decline as quickly as expected. We therefore assume that the subjects were basically more activated in the second neutral phase than in the first.

We also found a pattern in $O_2Hb$ and tHb levels indicating that prefrontal brain activity increased with repeated contact to the dog while it did not increase with repeated contact in the plush-animal condition. There seems to be a difference, especially between the first and the second contact with the dog suggesting that familiarity might play a different role in interactions with live and plush animals. However, the other two outcomes (HHb and oxygen saturation) did not show an increase with repeated contact and do not support this hypothesis. This result of this explorative analysis therefore needs to be further investigated in future studies.

## 4.3 Hypothesis about underlying mechanisms

We have different hypotheses explaining our result of higher activation in the dog condition compared to the plush-animal condition. The prefrontal cortex is known to be involved not only in executive functions such as attention control, working memory, and problem-solving but also in social and emotional processes [50, 51]. It has reciprocal connections with brain regions that are involved in emotional processing such as the amygdala and higher-order sensory regions within the temporal cortex [51].

Social interactions with animals are highly emotionally relevant for a majority of people [7, 31–34]. We thus hypothesize that interacting with the dog led to higher emotional involvement in the participants compared to interacting with the plush animal. This higher emotional involvement correlates with higher frontal activity. Previous studies using neuroimaging or behavioral outcomes support this hypothesis of higher emotional arousal by live animals [21, 39, 52–55].

Potential higher emotional involvement might in parallel also lead to more attention for and a stronger focus on the dog compared to the plush animal. Several authors have shown that interactions with animals can promote attention and activate attention networks [20, 21, 26, 56, 57]. Attentional processes such as attentional set-shifting or attention monitoring are located in the frontal cortex [50, 58, 59].

Another consequence of higher emotional arousal or of touching a live dog can be increased physiological arousal [60]. This arousal can be related to a positive state, but interacting with a dog could also cause higher stress than interacting with a plush animal. Further parameters such as heart rate or skin conductance are needed to distinguish physiological arousal from other processes such as emotional involvement and attention. Further, the increase in activation might also have been caused by a greater cognitive load as a dog is a more complex stimulus than a plush animal [61, 62]. A last hypothesis might be that motor control played a role [63, 64] as stroking a live dog might demand different motor adaption in the participants.

In sum, there are several possible explanations for our results that would benefit from being investigated in the future. Based on the recent literature, we hypothesize that emotional involvement might be a central underlying mechanism of the neurological frontal brain correlates of human–animal interaction. We therefore suppose that the increase in brain activity in the dog condition over the three contacts might be explained based on a developing relationship between the participant and the dog. Familiarity and a relationship with the dog could have raised the salience of the dog, kept the participant's attention on the dog's behavior, and increased emotional arousal during the experiment. An fMRI study on pet attachment found a correlation between pet attachment and brain activity in areas involved in increasing attention and attentional load [21].

## 4.4 Implications for clinical practice

It is important that future research tries to replicate our findings because they could have important implications for clinical practice such as animal-assisted therapy. Our results indicate that interactions with a dog might activate more attentional processes and elicit stronger emotional arousal than comparable nonliving stimuli. Moreover, it seems that especially close and active physical contact to a familiar dog might promote social attention in humans. This is especially relevant for patients with deficits in motivation, attention, and socioemotional functioning. High involvement is a crucial factor for learning, as has been shown in several studies [65, 66]. For example, it has been shown that emotional relevance is central [67].

If patients with deficits in motivation, attention, and socioemotional functioning show higher emotional involvement in activities connected to a dog, then such activities could increase the chance of learning and of achieving therapeutic aims. These hypotheses should be investigated in future studies, as they suggest that integrating animals into therapeutic interventions might be a promising approach for improving emotional involvement and attention.

## 4.5 Limitations and strengths

Blinding was not possible due to the nature of the study. Moreover, randomizing the sequence of the conditions was not completely possible because of the irregular presence of the dogs. It should also be noted that there was an additional person present during the presence of the dog. The dog owners did not interact with the participants during the measurements, but participants could see the dog owners in two-thirds of the sessions. For most of the outcomes, visibility of the dog owner had no effect, but this factor should be controlled in future studies. Moreover, we did not assess attitudes toward animals. The sample size reflects the pilot character of the study. The results thus must be interpreted carefully.

While fNIRS technology has several advantages, measurements of regional cerebral oxygen saturation can be affected by skull thickness, gyration, hemoglobin concentration, or extracranial blood flow [68, 69]. We decided to use fNIRS because it allowed the study to take place in a natural environment and did not produce any sounds that could irritate the participants or

the dogs. Since we repeatedly measured the outcomes for each condition and had a within-subject design where each participant served as their own control, these issues are limited. In addition, the probe design with a multidistance approach naturally reduces sensitivity to extra-cranial effects [70]. Drifts are also not likely because the fNIRS device corrects for that. Further, $O_2Hb$ concentration and oxygen saturation show the same pattern, which would not be the case if there was a drift. It could be argued that we should have detrended for the difference from the first to the second neutral phase. But the carry-over effect in the second neutral phase is not the same in the dog condition and the plush-animal condition. Detrending could thus have covered up effects that we assume reflect real changes.

The strengths of the study are that we investigated the effects of live dogs on neuronal activation instead of dogs presented via photos or videos and that we controlled for different levels of closeness and physical contact between the participant and the dog or the plush animal. We also carefully controlled the environmental factors in the room, the wording of the instructions, and the time of day of the sessions. Interactions between participants and the dog or the plush animal were standardized and kept as similar as possible. With regard to the plush animal used in the control condition, we controlled for tactile inputs such as its fur, warmth, and weight, and it was named and called by a name just as the dogs were called by a name in the study.

## 4.6 Future research

Future studies should take into account participants' characteristics like gender, pet ownership, and attitude toward animals. It has been shown that participants who loved horses exhibited lateralization while petting a horse. In contrast, participants who only "kind of liked" horses did not exhibit lateralization [24]. A study on brain activity during cat petting indicated a gender difference [25], and in an fMRI study, pet owners showed greater activation than non-pet owners while looking at images of unfamiliar pets [21]. Future research should replicate our findings with larger sample sizes and different participants. Moreover, the effects of direct interaction with a live dog could be investigated with other neuroimaging techniques that can measure brain activity in different brain areas simultaneously. It is important to further understand the effect of familiarity and relationship as well as of the type of interaction with the dog. To do so, future studies could use different interactions such as speaking to the dog or include reciprocal interactions such as playing with the dog. Familiarity and relationship should be systematically controlled by involving unfamiliar dogs, unfamiliar dogs with repeated contact, and participants' own pet dogs. It would be interesting to compare the effects of different animal species or of different features of dogs' appearances and to use different control conditions. Obtaining subjective ratings of the different interactions such as perceived pleasantness, stress, and relationship with the dog or plush animal should be introduced in the future. Moreover, imposing a concurrent cognitive task might be useful to see if the presence of a real dog has facilitating effects on behavioral performance. Moreover, it is important to test our hypotheses regarding clinical relevance. Future studies should involve patients with deficits in motivation, attention, and socioemotional functioning and investigate if the same results can be found regarding brain activity and also look at therapeutic outcomes such as achieving rehabilitation goals.

With regard to standardization, we recommend implementing a manipulation test to check for motor functions, to randomize the phases, and to control for the number of people in the room, the position of the dog owner, and the handedness of the participants. If it is possible, we would recommend implementing a pretask and posttask baseline. The length of the neutral phase should be longer to avoid carry-over effects.

## 5 Conclusion

The present study demonstrates that prefrontal brain activity in healthy subjects increased with a rise in interactional closeness with a dog or a plush animal. Moreover, participants had higher brain activation in the presence of a dog compared to in the presence of a plush animal. This indicates that interactions with a dog might activate more attentional processes and elicit stronger emotional arousal than comparable nonliving stimuli. Our results also suggest that a relationship with the dog might be a crucial factor. The results are clinically relevant for patients with deficits in motivation, attention, and socioemotional functioning. Integrating animals into therapeutic interventions might therefore be a promising approach for improving emotional involvement and attention.

## Acknowledgments

We would like to thank Daniel Ostojic for his advice on collecting and interpreting the fNIRS data. A great thanks goes to all the students who helped with data collection and preparation. We especially thank Felicitas Theis, Christina Zimmer, and Sabine Probst for their participation with their dogs Emma, Winnie, and Perla.

## Author Contributions

**Conceptualization:** Valentine L. Marcar, Margret Hund-Georgiadis, Karin Hediger.

**Data curation:** Rahel Marti, Valentine L. Marcar, Jan Hattendorf.

**Formal analysis:** Rahel Marti, Jan Hattendorf, Karin Hediger.

**Funding acquisition:** Karin Hediger.

**Investigation:** Rahel Marti, Milena Petignat.

**Methodology:** Rahel Marti, Milena Petignat, Valentine L. Marcar, Martin Wolf.

**Project administration:** Rahel Marti, Milena Petignat.

**Resources:** Martin Wolf, Margret Hund-Georgiadis.

**Software:** Martin Wolf.

**Supervision:** Valentine L. Marcar, Martin Wolf, Karin Hediger.

**Visualization:** Rahel Marti.

**Writing – original draft:** Rahel Marti.

**Writing – review & editing:** Milena Petignat, Valentine L. Marcar, Jan Hattendorf, Martin Wolf, Margret Hund-Georgiadis, Karin Hediger.

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
