## [Decision Letter · Decision Letter 0]

15 Jul 2022

PONE-D-22-10049Effects of contact with a dog on prefrontal brain activity: A controlled trialPLOS ONE

Dear Dr. Marti,

Thank you for submitting your manuscript to PLOS ONE. After careful consideration, we feel that it has merit but does not fully meet PLOS ONE’s publication criteria as it currently stands. Therefore, we invite you to submit a revised version of the manuscript that addresses the points raised during the review process. The reviewers' feedback is below. Please submit your revised manuscript by Aug 29 2022 11:59PM. If you will need more time than this to complete your revisions, please reply to this message or contact the journal office at plosone@plos.org. Please include the following items when submitting your revised manuscript:A rebuttal letter that responds to each point raised by the academic editor and reviewer(s). You should upload this letter as a separate file labeled 'Response to Reviewers'.A marked-up copy of your manuscript that highlights changes made to the original version. You should upload this as a separate file labeled 'Revised Manuscript with Track Changes'.An unmarked version of your revised paper without tracked changes. You should upload this as a separate file labeled 'Manuscript'.If applicable, we recommend that you deposit your laboratory protocols in protocols.io to enhance the reproducibility of your results. Protocols.io assigns your protocol its own identifier (DOI) so that it can be cited independently in the future. For instructions see: https://journals.plos.org/plosone/s/submission-guidelines#loc-laboratory-protocols. Additionally, PLOS ONE offers an option for publishing peer-reviewed Lab Protocol articles, which describe protocols hosted on protocols.io. Read more information on sharing protocols at https://plos.org/protocols?utm_medium=editorial-email&utm_source=authorletters&utm_campaign=protocols.

We look forward to receiving your revised manuscript.

Kind regards,

Anna Manelis, Ph.D.

Academic Editor

PLOS ONE

Journal Requirements:

Reviewers' comments:

Reviewer's Responses to Questions

**Comments to the Author**

1. Is the manuscript technically sound, and do the data support the conclusions?

Reviewer #1: Yes

Reviewer #2: Partly

2. Has the statistical analysis been performed appropriately and rigorously? 

Reviewer #1: Yes

Reviewer #2: Yes

3. Have the authors made all data underlying the findings in their manuscript fully available?

Reviewer #1: Yes

Reviewer #2: No

4. Is the manuscript presented in an intelligible fashion and written in standard English?

Reviewer #1: Yes

Reviewer #2: Yes

5. Review Comments to the Author

Reviewer #1: The abstract should be more structured it is written rather casually. There are several areas in the paper where they are also casual in their writing such as this in the methods "We opted fNIRS to measure..." It should not come across as a random selection of technology It should be a scientifically rational and intentional selection of measurement.

Line 52 pg. 1 uses the term neurochemicals it should be neurotransmitters

Methods In the methods section line 99-101 the design is described as following protocol for subjects are described as patients with severe disorder of consciousness this needs to be clarified it is very unclear what this means. They then go on to describe healthy participants. Please clarify why it is relevant to follow a method for people with sever disorder of consciousness.

The authors need t provide more specific information about the brain regions measured with fNIRS it simply states The study staff attached two fNIRS sensors to measure oxygen saturation on the participants’ foreheads. They also refer to This corresponded to locations F1, F3, F2 and F4 on the frontopolar area what Broca brain regions are these?

Please spell check the paper carefully in the first paragraph of results it says "planed sessions" it should be planned sessions.

An explanation of lateralization is needed so readers can understand why it is important to consider it is introduced in section 4.1 without background.

The hypotheses in the discussion session are clearer than the hypotheses stated in the introduction. Please revise them to be clearer currently it is wrotten as 'First, we expected the degree of closeness in contact with a

88 dog or plush animal to be correlated with the amount of stimulation and therefore also with

89 brain activity. Second, we hypothesized that.."

Perhaps this paper would provide an example of the level of detail needed Bergen-Cico, D., Grant, T. et al. (2021). Using fNIRS to Examine Neural Mechanisms of Change Associated with Mindfulness-Based Interventions for Stress and Trauma: Results of a Pilot Study for Women. Mindfulness, 12(9), 2295-2310.

Reviewer #2: The study aimed to know effects of animal contact on human mental and physical health by investigating changes in frontal brain activity. The manuscript is well written and most of the details were well described. The statistical methods were well strong and well explained. However there are some suggestions that should be considered in a revised manuscript.

(1) The authors should give more details of how to determine the sample size. In my understanding, if the sample size (n=21) is not enough, then the authors should discuss this limitation deeply in the discussion section.

(2) There are some factors that may influence the interaction effects, such as the social-emotional competence of participants, and the features of dogs’ face (eg. cute VS vicious), among others. The authors should discuss a little bit about this points.

(3) The authors should discuss deeply the difference of prefrontal brain activity between different contracts with dogs.

6. PLOS authors have the option to publish the peer review history of their article (what does this mean?). If published, this will include your full peer review and any attached files.

Reviewer #1: **Yes: **Dessa Bergen-Cico

Reviewer #2: No

---

## [Author Response · Author response to Decision Letter 0]

29 Aug 2022

Dear Prof. Dr. Manelis

We would like to resubmit our manuscript entitled “Effects of Contact with a Dog on Prefrontal Brain Activity: A Controlled Trial" after responding to all the comments of the reviewers and changing our manuscript accordingly. We uploaded two versions of our manuscript, a clean version without track changes and a version where all changes are visible. 

As requested, this cover letter includes a point-by point reply to each criticism of the referees and a detailed indication of the changes made in our manuscript. All our responses are in navy blue. 

We checked and corrected the format of the reference section as requested. Further, we saw that the figure references in the text were inconsistent. We therefore modified them.

We changed our Data Availability statement and provided a link to get access to our data file (see track changes on page 23, lines 535–536:

“Data are available from the Harvard Dataverse database at https://doi.org/10.7910/DVN/U57AK0.”

We hope you find our manuscript suitable for publication and we look forward to hearing from you in due course.

Sincerely, 

M.Sc. Rahel Marti

Assistant

 

Reviewer #1

The abstract should be more structured it is written rather casually. 

Thank you very much for your comment. We checked the language and changed it accordingly. Further, we structured the abstract, added more detailed information about the study design and added a paragraph on the study’s findings. See track changes on page 2, lines 30-51:

“Background: There is a broad range of known effects of animal contact on human mental and physical health. Neurological correlates of human interaction with animals have been sparsely investigated. We investigated changes in frontal brain activity in the presence of and during contact with a dog. 

Methods: Twenty-one healthy individuals each participated in six sessions. In three sessions, participants had contact with a dog, and in three control sessions they interacted with a plush animal. Each session had five two-minute phases with increasing intensity of contact to the dog or plush animal from the first to the fourth phase. We measured oxygenated, deoxygenated, and total hemoglobin and oxygen saturation of the blood in the frontal lobe/frontopolar area with functional near-infrared spectroscopy (SenSmart Model X-100) to assess brain activity. 

Findings: In both conditions, the concentration of oxygenated hemoglobin increased significantly from the first to the fourth phase by 2.78 �mol/l (CI = 2.03–3.53, p < .001). Oxygenated hemoglobin concentration was 0.80 �mol/l higher in the dog condition compared to in the control condition (CI = 0.27–1.33, p = .004). Deoxygenated-hemoglobin concentration, total hemoglobin concentration, and oxygen saturation showed similar patterns. 

Conclusion: Prefrontal brain activation in healthy subjects increased with the rise in interaction closeness with a dog or a plush animal. Moreover, interaction with a dog stimulated more brain activity compared to the control condition, suggesting that interactions with a dog can activate stronger attentional processes and elicit more emotional arousal than interacting with a nonliving stimulus.” 

There are several areas in the paper where they are also casual in their writing such as this in the methods "We opted fNIRS to measure..." It should not come across as a random selection of technology It should be a scientifically rational and intentional selection of measurement.

Thank you for this helpful comment. We changed the term and let the manuscript be checked by a professional language proofreader who checked the text with a distinct focus on avoiding casual language:

“We chose fNIRS to measure the response in the prefrontal cortex as it is particularly suited for investigating the neuronal correlates of such a complex social situation of human–animal interaction.” (See track changes on page 7, lines 176–178)

To see further adjustments made by the professional language proofreader, you can look at the track changes in the text. 

Line 52 pg. 1 uses the term neurochemicals it should be neurotransmitters

Thank you very much for your comment and your suggestion. �-endorphin and oxytocin are, as you wrote, neuropeptides and neurotransmitters whereas prolactin is a hormone not functioning as a neurotransmitter. Therefore, we chose the word neurochemicals as an umbrella term to summarize all three.

Methods In the methods section line 99-101 the design is described as following protocol for subjects are described as patients with severe disorder of consciousness this needs to be clarified it is very unclear what this means. They then go on to describe healthy participants. Please clarify why it is relevant to follow a method for people with sever disorder of consciousness.

We thank you for pointing out this important issue. The way we wrote it was indeed unclear to the reader. We changed the locating of this sentence to give it more context and added more information regarding a planned second study with patients with severe disorders of consciousness:

“We planned to compare the results of this study with a study population of patients with severe disorders of consciousness in a future trial, so the study design complied with the requirements for measuring a group of patients with severe disorders of consciousness.” (See track changes on pages 5, lines 114–117)

The authors need t provide more specific information about the brain regions measured with fNIRS it simply states The study staff attached two fNIRS sensors to measure oxygen saturation on the participants’ foreheads. They also refer to This corresponded to locations F1, F3, F2 and F4 on the frontopolar area what Broca brain regions are these?

Thank you for your comment. The Broca area is a distinct area corresponding to Brodmann areas 44 and 45 and we did not measure on Broca’s area. To answer this concern, we added the corresponding Brodmann areas for the prefrontal cortex which are number 9, 10, and 46, see page 8, lines 193–195:

“This corresponded to locations F1, F3, F2, and F4 on the frontopolar area according to the international 10–20 system and to the Brodmann areas 9, 10, and 46.”

Please spell check the paper carefully in the first paragraph of results it says "planed sessions" it should be planned sessions.

Thank you for finding this spelling error and pointing this out. We changed this and, as stated above, checked the manuscript by a professional proofreader:

“We conducted 119 of the 126 planned sessions (Fig 2).” (See track changes on page 10, line 231)

An explanation of lateralization is needed so readers can understand why it is important to consider it is introduced in section 4.1 without background.

Thank you very much for your comment and your suggestion. Indeed, it was not introduced nicely so it was hard to understand for the readers. Based on your suggestion we changed the order of the text so that it is now embedded in the context of findings of other studies to make it clear why lateralization is important. See track changes on page 17, lines 372–378: 

“Other studies identified lateralized activation patterns in frontal areas during petting a horse or a cat compared to a plush animal [24, 25]. For example, participants exhibited lateralization in the right frontopolar cortex while petting a real horse compared to no lateralization while petting a plush horse [24]. The authors attributed the lateralized activity to differences in function of the left and right frontal regions. We did not test for lateralization in the present study, but visual inspection of our data does not suggest lateralization. However, future studies should address the possibility of lateralization.”

The hypotheses in the discussion session are clearer than the hypotheses stated in the introduction. Please revise them to be clearer currently it is wrotten as 'First, we expected the degree of closeness in contact with a

88 dog or plush animal to be correlated with the amount of stimulation and therefore also with

89 brain activity. Second, we hypothesized that.."

We thank you for pointing this out. We adapted the wording of the first hypothesis: 

“We expected, first, that the increase of closeness in contact with a dog or plush animal would correlate with an increased amount of stimulation and therefore also with increased brain activity.” (See track changes on page 4, lines 95–97)

Perhaps this paper would provide an example of the level of detail needed Bergen-Cico, D., Grant, T. et al. (2021). Using fNIRS to Examine Neural Mechanisms of Change Associated with Mindfulness-Based Interventions for Stress and Trauma: Results of a Pilot Study for Women. Mindfulness, 12(9), 2295-2310.

Thank you for your comment and direct our attention to this study. We added a new figure to make extend the description of the procedures and make it more accessible for the readers. 

See Figure 1, on page 6, line 155. 

Reviewer #2

(1) The authors should give more details of how to determine the sample size. In my understanding, if the sample size (n=21) is not enough, then the authors should discuss this limitation deeply in the discussion section.

Thank you very much for your comment and your suggestion. Since the follow-up project of this pilot study aimed at conducting the same procedure with patients with severe disorders of consciousness, the sample size was determined based on feasibility regarding the number of patients with severe disorders of consciousness in this hospital as well as the results of a previous study. In this study, we analyzed behavioral outcomes from 19 patients with acquired brain injury and found a significant effect of animal-assisted therapy on positive emotions and social behavior and a piloting trial with the same design where we analyzed four participants [1].

The power of the current study was adequate as it was possible to find differences between conditions. However, since this was a pilot study, we mentioned this fact and the sample size in the limitations section: 

“The sample size reflects the pilot character of the study. The results thus must be interpreted carefully.“ (See track changes on page20, line 457-459) 

(2) There are some factors that may influence the interaction effects, such as the social-emotional competence of participants, and the features of dogs’ face (eg. cute VS vicious), among others. The authors should discuss a little bit about this points.

Thank you very much for your comment and your suggestion. Indeed, socio-emotional competencies of participants and features of the dogs play a role in the interaction between humans and dogs. However, in our study design, these factors cannot influence the results because we compared the participants with themselves (within-subject design). But we agree with you that it would be important for future research to investigate more closely what characteristics of the human participants and of the dogs predict different reactions within such a human-animal interaction. 

In lines 503–505 on page 22 we pointed out, that future trials should investigate participants with varying amount of socio-emotional competences. Based on your suggestion, we also added on page 22 in lines 496–498: 

“It would be interesting to compare the effects of different animal species or of different features of dogs’ appearances and to use different control conditions.”

(3) The authors should discuss deeply the difference of prefrontal brain activity between different contracts with dogs.

Thank you for your comment. We extended the discussion about the effects of the increasing intensity of contact and added the following text: (page 16, line 352–360): 

“We found that prefrontal brain activity increased with a rise in the intensity of contact with a dog or a plush animal. From watching the animal to feeling it passively to actively petting the animal, the interactional closeness increased and, with it the intensity of stimulation as well as the number of senses involved. This led to an increase in brain activation. We detected the same pattern in a pilot study with a similar study design and comparable forms of contact to an animal [27]. In line with this, another study revealed higher frontopolar activity when participants stroked a plush animal or a miniature horse compared to just seeing them [24]. Moreover, stroking a cat stimulated higher activation of the inferior frontal gyrus compared to just touching a cat [25].”

1. Arnskötter W, Marcar VL, Wolf M, Hund-Georgiadis M, Hediger K. Animal presence modulates frontal brain activity of patients in a minimally conscious state: A pilot study. Neuropsychol Rehabil 2021:1–13. doi:10.1080/09602011.2021.1886119.

---

## [Editor Report · Decision Letter 1]

5 Sep 2022

Effects of contact with a dog on prefrontal brain activity: A controlled trial

PONE-D-22-10049R1

Dear Dr. Marti,

We’re pleased to inform you that your manuscript has been judged scientifically suitable for publication and will be formally accepted for publication once it meets all outstanding technical requirements.

Kind regards,

Anna Manelis, Ph.D.

Academic Editor

PLOS ONE
---

## [Editor Report · Acceptance letter]

12 Sep 2022

PONE-D-22-10049R1 

Effects of contact with a dog on prefrontal brain activity: A controlled trial 

Dear Dr. Marti:

I'm pleased to inform you that your manuscript has been deemed suitable for publication in PLOS ONE. Congratulations! Your manuscript is now with our production department. 

Kind regards, 

on behalf of

Dr. Anna Manelis 

Academic Editor

PLOS ONE